# Immediate Psychological Responses and Associated Factors during the Initial Stage of the 2019 Coronavirus Disease (COVID-19) Epidemic among the General Population in China

**DOI:** 10.3390/ijerph17051729

**Published:** 2020-03-06

**Authors:** Cuiyan Wang, Riyu Pan, Xiaoyang Wan, Yilin Tan, Linkang Xu, Cyrus S. Ho, Roger C. Ho

**Affiliations:** 1Institute of Cognitive Neuroscience, Faculty of Education, Huaibei Normal University, Huaibei 235000, China; wcy@chnu.edu.cn (C.W.); riyu0402@chnu.edu.cn (R.P.); wming624@sina.com (X.W.); 977367tan@sina.com (Y.T.); lgb@chnu.edu.cn (L.X.); 2Department of Psychological Medicine, National University Health System, Kent Ridge 119228, Singapore; su_hui_ho@nuhs.edu.sg; 3Department of Psychological Medicine, Yong Loo Lin School of Medicine, National University of Singapore, Kent Ridge 119228, Singapore; 4Institute of Health Innovation and Technology (iHealthtech), National University of Singapore, Kent Ridge 119077, Singapore

**Keywords:** anxiety, coronavirus, depression, epidemic, knowledge, precaution, psychological impact, respiratory symptoms, stress

## Abstract

*Background:* The 2019 coronavirus disease (COVID-19) epidemic is a public health emergency of international concern and poses a challenge to psychological resilience. Research data are needed to develop evidence-driven strategies to reduce adverse psychological impacts and psychiatric symptoms during the epidemic. The aim of this study was to survey the general public in China to better understand their levels of psychological impact, anxiety, depression, and stress during the initial stage of the COVID-19 outbreak. The data will be used for future reference. *Methods:* From 31 January to 2 February 2020, we conducted an online survey using snowball sampling techniques. The online survey collected information on demographic data, physical symptoms in the past 14 days, contact history with COVID-19, knowledge and concerns about COVID-19, precautionary measures against COVID-19, and additional information required with respect to COVID-19. Psychological impact was assessed by the Impact of Event Scale-Revised (IES-R), and mental health status was assessed by the Depression, Anxiety and Stress Scale (DASS-21). *Results:* This study included 1210 respondents from 194 cities in China. In total, 53.8% of respondents rated the psychological impact of the outbreak as moderate or severe; 16.5% reported moderate to severe depressive symptoms; 28.8% reported moderate to severe anxiety symptoms; and 8.1% reported moderate to severe stress levels. Most respondents spent 20–24 h per day at home (84.7%); were worried about their family members contracting COVID-19 (75.2%); and were satisfied with the amount of health information available (75.1%). Female gender, student status, specific physical symptoms (e.g., myalgia, dizziness, coryza), and poor self-rated health status were significantly associated with a greater psychological impact of the outbreak and higher levels of stress, anxiety, and depression (*p* < 0.05). Specific up-to-date and accurate health information (e.g., treatment, local outbreak situation) and particular precautionary measures (e.g., hand hygiene, wearing a mask) were associated with a lower psychological impact of the outbreak and lower levels of stress, anxiety, and depression (*p* < 0.05). *Conclusions:* During the initial phase of the COVID-19 outbreak in China, more than half of the respondents rated the psychological impact as moderate-to-severe, and about one-third reported moderate-to-severe anxiety. Our findings identify factors associated with a lower level of psychological impact and better mental health status that can be used to formulate psychological interventions to improve the mental health of vulnerable groups during the COVID-19 epidemic.

## 1. Introduction

The 2019 coronavirus disease (COVID-19) epidemic in China is a global health threat [1], and is by far the largest outbreak of atypical pneumonia since the severe acute respiratory syndrome (SARS) outbreak in 2003. Within weeks of the initial outbreak the total number of cases and deaths exceeded those of SARS [2]. The outbreak was first revealed in late December 2019 when clusters of pneumonia cases of unknown etiology were found to be associated with epidemiologically linked exposure to a seafood market and untraced exposures in the city of Wuhan of Hubei Province [3]. Since then, the number of cases has continued to escalate exponentially within and beyond Wuhan, spreading to all 34 regions of China by 30 January 2020. On the same day, the World Health Organization (WHO) declared the COVID-19 outbreak a public health emergency of international concern [4].

COVID-19, similarly to SARS, is a beta-coronavirus that can be spread to humans through intermediate hosts such as bats [5], though the actual route of transmission is still debatable. Human-to-human transmission has been observed via virus-laden respiratory droplets, as a growing number of patients reportedly did not have animal market exposure, and cases have also occurred in healthcare workers [6]. Transmissibility of COVID-19 as indicated by its reproductive number has been estimated at 4.08 [7], suggesting that on average, every case of COVID-19 will create up to 4 new cases. The reporting rate after 17 January 2020 has been considered to have increased 21-fold in comparison to the situation in the first half of January 2020 [8]. The average incubation period is estimated to be 5.2 days, with significant variation among patients [9], and it may be capable of asymptomatic spread [10,11]. Symptoms of infection include fever, chills, cough, coryza, sore throat, breathing difficulty, myalgia, nausea, vomiting, and diarrhea [12]. Older men with medical comorbidities are more likely to get infected, with worse outcomes [12]. Severe cases can lead to cardiac injury, respiratory failure, acute respiratory distress syndrome, and death [13]. The provisional case fatality rate by WHO is around 2%, but some researchers estimate the rate to range from 0.3% to 0.6% [14].

Since the outbreak, response efforts by the China government have been swift, and three weeks into the epidemic, in an unprecedented move to retard the spread of the virus, a lockdown was imposed on Wuhan on 23 January, with travel restrictions. Within days, the quarantine was extended to additional provinces and cities, affecting more than 50 million people in total. Many stayed at home and socially isolated themselves to prevent being infected, leading to a “desperate plea” [15]. There have also been accounts of shortages of masks and health equipment. The ongoing COVID-19 epidemic is inducing fear, and a timely understanding of mental health status is urgently needed for society [16]. Previous research has revealed a profound and wide range of psychosocial impacts on people at the individual, community, and international levels during outbreaks of infection. On an individual level, people are likely to experience fear of falling sick or dying themselves, feelings of helplessness, and stigma [17]. During one influenza outbreak, around 10% to 30% of general public were very or fairly worried about the possibility of contracting the virus [18]. With the closure of schools and business, negative emotions experienced by individuals are compounded [19]. During the SARS outbreak, many studies investigated the psychological impact on the non-infected community, revealing significant psychiatric morbidities which were found to be associated with younger age and increased self-blame [20]. Those who were older, of female gender, more highly educated, with higher risk perceptions of SARS, a moderate anxiety level, a positive contact history, and those with SARS-like symptoms were more likely to take precautionary measures against the infection [21].

Currently, there is no known information on the psychological impact and mental health of the general public during the peak of the COVID-19 epidemic. This is especially pertinent with the uncertainty surrounding an outbreak of such unparalleled magnitude. Based on our understanding, most of the research related to this outbreak focuses on identifying the epidemiology and clinical characteristics of infected patients [6,12], the genomic characterization of the virus [22], and challenges for global health governance [23]. However, there are no research articles examining the psychological impact on COVID-19 on the general population in China.

Therefore, this present study represents the first psychological impact and mental health survey conducted in the general population in China within the first two weeks of the COVID-19 outbreak. This study aims to establish the prevalence of psychiatric symptoms and identify risk and protective factors contributing to psychological stress. This may assist government agencies and healthcare professionals in safeguarding the psychological wellbeing of the community in the face of COVID-19 outbreak expansion in China and different parts of the world.

## 2. Methods

### 2.1. Setting and Participants

We adopted a cross-sectional survey design to assess the public’s immediate psychological response during the epidemic of COVID-19 by using an anonymous online questionnaire. A snowball sampling strategy, focused on recruiting the general public living in mainland China during the epidemic of COVID-19, was utilized. The online survey was first disseminated to university students and they were encouraged to pass it on to others.

### 2.2. Procedure

As the Chinese Government recommended the public to minimize face-to-face interaction and isolate themselves at home, potential respondents were electronically invited by existing study respondents. They completed the questionnaires in Chinese through an online survey platform (‘SurveyStar’, Changsha Ranxing Science and Technology, Shanghai, China). Expedited ethics approval was obtained from the Institutional Review Board of the Huaibei Normal University (HBU-IRB-2020-001), which conformed to the principles embodied in the Declaration of Helsinki. Information about this study was posted on a dedicated university website. All respondents provided informed consent. Data collection took place over three days (31 January–2 February 2020) after the WHO declared the COVID-19 outbreak as a public health emergency of international concern.

### 2.3. Survey Development

Previous surveys on the psychological impacts of SARS and influenza outbreaks were reviewed [18,21,24]. Authors included additional questions related to the COVID-19 outbreak. The structured questionnaire consisted of questions that covered several areas: (1) demographic data; (2) physical symptoms in the past 14 days; (3) contact history with COVID-19 in the past 14 days; (4) knowledge and concerns about COVID-19; (5) precautionary measures against COVID-19 in the past 14 days; (6) additional information required with respect to COVID-19; (7) the psychological impact of the COVID-19 outbreak; and (8) mental health status.

Sociodemographic data were collected on gender, age, education, residential location in the past 14 days, marital status, employment status, monthly income, parental status, and household size. Physical symptom variables in the past 14 days included fever, chills, headache, myalgia, cough, difficulty in breathing, dizziness, coryza, sore throat, and persistent fever, as well as persistent fever and cough or difficulty breathing. Respondents were asked to rate their physical health status and state any history of chronic medical illness. Health service utilization variables in the past 14 days included consultation with a doctor in the clinic, admission to the hospital, being quarantined by a health authority, and being tested for COVID-19. Contact history variables included close contact with an individual with confirmed COVID-19, indirect contact with an individual with confirmed COVID-19, and contact with an individual with suspected COVID-19 or infected materials.

Knowledge about COVID-19 variables included knowledge about the routes of transmission, level of confidence in diagnosis, level of satisfaction of health information about COVID-19, the trend of new cases and death, and potential treatment for COVID-19 infection. Respondents were asked to indicate their source of information. The actual number of confirmed cases of COVID-19 and deaths in the city on the day of the survey were collected. Concern about COVID-19 variables included self and other family members contracting COVID-19 and the chance of surviving if infected.

Precautionary measures against COVID-19 variables included avoidance of sharing of utensils (e.g., chopsticks) during meals, covering mouth when coughing and sneezing, washing hands with soap, washing hands immediately after coughing, sneezing, or rubbing the nose, washing hands after touching contaminated objects, and wearing a mask regardless of the presence or absence of symptoms. The respondents were asked the average number of hours staying at home per day to avoid COVID-19. Respondents were also asked whether they felt too much -unnecessary worry had been made about the COVID-19 epidemic. Additional health information about COVID-19 needed by respondents included more information about symptoms after contraction of COVID-19, routes of transmission, treatment, prevention of the spread of COVID-19, local outbreaks, travel advice, and other measures imposed by other countries.

The psychological impact of COVID-19 was measured using the Impact of Event Scale-Revised (IES-R). The IES-R is a self-administered questionnaire that has been well-validated in the Chinese population for determining the extent of psychological impact after exposure to a public health crisis within one week of exposure [25]. This 22-item questionnaire is composed of three subscales and aims to measure the mean avoidance, intrusion, and hyperarousal [26]. The total IES-R score was divided into 0–23 (normal), 24–32 (mild psychological impact), 33–36 (moderate psychological impact), and >37 (severe psychological impact) [27].

Mental health status was measured using the Depression, Anxiety and Stress Scale (DASS-21) and calculations of scores were based on the previous study [28]. Questions 3, 5, 10, 13, 16, 17 and 21formed the depression subscale. The total depression subscale score was divided into normal (0–9), mild depression (10–12), moderate depression (13–20), severe depression (21–27), and extremely severe depression (28–42). Questions 2, 4, 7, 9, 15, 19, and 20 formed the anxiety subscale. The total anxiety subscale score was divided into normal (0–6), mild anxiety (7–9), moderate anxiety (10–14), severe anxiety (15–19), and extremely severe anxiety (20–42). Questions 1, 6, 8, 11, 12, 14, and 18 formed the stress subscale. The total stress subscale score was divided into normal (0–10), mild stress (11–18), moderate stress (19–26), severe stress (27–34), and extremely severe stress (35–42). The DASS has been demonstrated to be a reliable and valid measure in assessing mental health in the Chinese population [29,30]. The DASS was previously used in research related to SARS [31].

### 2.4. Statistical Analysis

Descriptive statistics were calculated for sociodemographic characteristics, physical symptoms and health service utilization variables, contact history variables, knowledge and concern-related variables, precautionary measure variables, and additional health information variables. Percentages of response were calculated according to the number of respondents per response with respect to the number of total responses of a question. The scores of the IES-R and DASS subscales were expressed as mean and standard deviation. We used linear regressions to calculate the univariate associations between sociodemographic characteristics, physical symptom and health service utilization variables, contact history variables, knowledge and concern variables, precautionary measure variables, additional health information variables, and the IES-S score as well as the subscales of the DASS. All tests were two-tailed, with a significance level of *p* < 0.05. Statistical analysis was performed using SPSS Statistic 21.0 (IBM SPSS Statistics, New York, United States).

## 3. Results

### 3.1. Development of the COVID-19 Epidemic from January 7 to February 2 2020

Figure 1 shows the development trend of the COVID-19 epidemic in China in January and February 2020. Since China first announced the national epidemic data on 20 January 2020, the number of confirmed cases, suspected cases, recovered individuals, and deaths related to COVID-19 infection have continued to escalate, with a sharp increase in the number of suspected cases after 26 January 2020. Both children and the elderly have been particularly vulnerable to the virus, with the youngest confirmed case being that of a 9-month-old infant.

### 3.2. Survey Respondents

We received responses from 1304 respondents, and 102 respondents did not complete the questionnaires. Eventually, we included 1210 respondents from 194 cities in China who had completed the questionnaires (completion rate: 92.79%). Overall, 1120 respondents submitted the questionnaires on the first day (31 January), 86 respondents submitted the questionnaires on the second day (1 February), and only 4 respondents submitted the questionnaires on the third day (2 February).

The psychological impact of COVID-19 outbreak, measured using the IES-R scale, revealed a sample mean score of 32.98 (SD = 15.42). Of all respondents, 296 (24.5%) reported minimal psychological impact (score < 23); 263 (21.7%) rated mild psychological impact (scores 24–32); and 651 (53.8%) reported a moderate or severe psychological impact (score > 33). Respondents’ depression, anxiety and stress levels, measured using the DASS 21-item scale, revealed a sample mean score of 20.16 (SD = 20.42). For the depression subscale, 843 (69.7%) were considered to have a normal score (score: 0–9); 167 (13.8%) were considered to suffer from mild depression (score: 10–12); 148 (12.2%) were considered to suffer from moderate depression (score: 13–20); and 52 (4.3%) were considered to suffer from severe and extremely severe depression (score: 21–42). For the anxiety subscale, 770 (63.6%) were considered to have a normal score (score: 0–6); 91 (7.5%) were considered to suffer from mild anxiety (score: 7–9); 247 (20.4%) were considered to suffer from moderate anxiety (score: 10–14); and 102 (8.4%) were considered to suffer from severe and extremely severe anxiety (score: 15–42). For the stress subscale, 821 (67.9%) were considered to have a normal score (score: 0–10); 292 (24.1%) were considered to suffer from mild stress (score: 11–18); 66 (5.5%) were considered to suffer from moderate stress (score: 19–26); and 31 (2.6%) were considered to suffer from severe and extremely severe stress (score: 27–42).

### 3.3. Sociodemographic Variables and Psychological Impact

Sociodemographic characteristics are presented in Table 1. The majority of respondents were women (67.3%), aged 21.4 to 30.8 years (53.1%), married (76.4%), with a household size of 3–5 people (80.7%), with children (67.4%), students (52.8%), and well educated (87.9% ≥ bachelor’s degree). Male gender was significantly associated with lower scores in the IES-R (B = −0.20, 95% Confidence Interval (95% CI) −0.35 to −0.05) but higher scores in the DASS stress subscale (B = 0.10, 95% CI: 0.02 to 0.19), DASS anxiety subscale (B = 0.19 95% CI: 0.05 to 0.33), and DASS depression subscale (B = 0.12, 95% CI: 0.01 to 0.23). Student status was significantly associated with higher IES-R (B = 0.20, 95% CI: 0.05 to 0.35), DASS stress subscale (B = 0.11, 95% CI: 0.02 to 0.19), and DASS anxiety subscale (B = 0.16, 95% CI: 0.02 to 0.30) scores as compared to those who were employed. Uneducated status was significantly associated with higher DASS depression subscale scores (B = 1.81, 95% CI: 0.46 to 3.16). Other sociodemographic variables including age, parental status, marital status, and household size were not associated with IES-R and DASS subscale scores.

### 3.4. Symptoms and Psychological Impact

For physical symptoms, Table 2 shows that 0.5% of the sample reported a fever of 38 °C for at least one day within the previous two weeks. Some respondents reported a range of physical symptoms, most frequently coryza (16.9%), cough (13.9%), sore throat (11.5%), headache (9.7%), myalgia (7.9%), dizziness (7.3%), chills (3.5%), fever (0.5%), and breathing difficulty (0.4%). Around 0.3% of respondents reported a dyad of symptoms such as fever with cough or fever with breathing difficulty. Overall, 793 respondents reported no symptoms (60.81%); 182 respondents reported one symptom (15.04%); 114 respondents reported two symptoms (9.42%); and 68 respondents reported three symptoms (5.62%). Linear regression showed that chills, myalgia, cough, dizziness, coryza, and sore throat were significantly associated with higher IES-R, DASS stress subscale, DASS anxiety subscale, and DASS depression subscale scores, while breathing difficulty was associated with only DASS anxiety and depression subscale scores. In contrast, the presence of a dyad of symptoms such as fever with cough or breathing difficulty was not associated with IES-R, DASS stress subscale, DASS anxiety subscale, and DASS depression subscale scores.

### 3.5. Health Status and Psychological Impact

In the prior two weeks, 3.5% of respondents had consulted a doctor in the clinic; 0.3% had been admitted to the hospital; 0.9% had been tested for COVID-19; 2.1% had been under quarantine by a health authority; and 68.3% reported good or very good health status. Around 93.6% of respondents did not suffer from any chronic illness, and 92.4% were covered by medical insurance. Clinic consultations (B = 0.38, 95% CI: 0.02 to 0.73) and hospitalizations (B = 1.23, 95 % CI: 0.09 to 2.36) were significantly associated with higher DASS anxiety subscale score. Poor or very poor self-rated health status was significantly associated with a greater psychological impact of the outbreak (B = 0.76, 95% CI: 0.02 to 1.49), and higher DASS stress subscale (B = 0.45, 95% CI: 0.02 to 0.88), DASS anxiety subscale (B = 0.90, 95% CI: 0.22 to 1.58), and DASS depression subscale (B = 0.65, 95% CI: 0.10 to 1.20) scores as compared to those with very good or good self-rated health status. History of chronic illness was significantly associated with higher IES-R, DASS stress subscale, DASS anxiety subscale, and DASS depression subscale scores.

### 3.6. Contact History and Psychological Impact

Table 3 shows the contact history of confirmed and suspected cases of COVID-19. Overall, 1% of respondents had been in contact with an individual with suspected COVID-19 or infected materials; 0.5% reported indirect contact with an individual with confirmed COVID-19; and 0.3% reported close contact with an individual with confirmed COVID-19. Variables in the contact history were not associated with IES-R and DASS scores, with the exception of contact with an individual with suspected COVID-19 or infected materials, which were significantly associated with anxiety (B = 0.98, 95% CI: 0.32 to 1.64).

### 3.7. Knowledge about COVID-19 and Psychological Impact

Regarding knowledge about COVID-19, Table 4 also shows that the most common perceived route of transmission was through droplets (92.1%), followed by contaminated objects (73.7%), and airborne transmission (60.5%). Nearly all respondents had heard that the number of infected individuals had increased (98.8%), the number of deaths had increased (97.8%), and the number of recovered individuals had increased (93.3%). The most common source of health information about COVID-19 was from the Internet (93.5%). The majority of respondents (75.1%) were very satisfied or fairly satisfied with the amount of health information available. Dissatisfaction with the amount of health information available about COVID-19 was significantly associated with higher IES-R score (B = 0.63, 95% CI: 0.11 to 1.14) and DASS stress subscale score (B = 0.32, 95% CI: 0.02 to 0.62). The information on the increase in the number of recovered individuals was significantly associated with a low DASS stress subscale score (B =−0.24, 95% CI: −0.40 to −0.07).

### 3.8. Concerns about COVID -19 and Psychological Impact

Regarding concerns about COVID-19, about 75.2% of respondents were very worried or somewhat worried about other family members getting COVID-19. In contrast, 50.9% of respondents were very worried or somewhat worried about a child younger than 16 years getting COVID-19. About 46.5% of the respondents expressed a high level of confidence in their doctor’s ability to diagnose or recognize COVID-19; and 46.1% believed the risk of contracting COVID-19 during the current outbreak was unlikely or not likely at all. The majority of respondents (69.2%) believed that they would be very likely or somewhat likely to survive COVID-19 if infected.

Those who had no confidence in their own doctor’s ability to diagnose or recognize COVID-19 were significantly more likely to have higher scores in the DASS stress subscale (B = 1.18, 95% CI: 0.61–1.75), DASS anxiety subscale (B = 1.86, 95% CI: 0.96 to 2.76), and DASS depression subscale (B = 1.66, 95% CI:0.94 to 2.38). A higher perceived likelihood of contracting COVID-19 during the current outbreak was significantly associated with lower IES-R score (B = −0.33, 95% CI: −0.61 to −0.05). In contrast, low perceived likelihood of contracting COVID-19 during the current outbreak was significantly associated with low DASS stress subscale (B = −0.18, 95% CI: −0.35 to −0.01) and low DASS anxiety subscale (B = −0.36, 95% CI: −0.63 to −0.09) scores. A low perceived likelihood of surviving COVID-19 if infected was significantly associated with high DASS stress subscale score (B = 0.34, 95% CI: 0.01 to 0.68).

High levels of concern about other family members getting COVID-19 were significantly associated with higher DASS stress subscale scores (B = 0.50, 95% CI: 0.04 to 0.96). Similarly, high levels of concern about a child younger than 16 years getting COVID-19 were significantly associated with higher IES-R scores (B = 0.25, 95% CI: 0.05 to 0.44) and DASS anxiety subscale scores (B = 0.24, 95% CI: 0.07 to 0.42).

### 3.9. Precautionary Measures and Psychological Impact

Table 5 shows the precautionary measures adopted by the respondents in the past 14 days, which were most frequently always washing hands after touching contaminated objects (66.6%), always wearing a mask regardless of the presence or absence of symptoms (59.8%), always covering mouth when coughing and sneezing (57.4%), always washing hands with soap (56.5%), always washing hands immediately after coughing, sneezing, or rubbing nose (41%), and always avoiding sharing utensils (e.g., chopsticks) during meals (40.5%). Linear regression analysis showed that avoiding the sharing of utensils (e.g., chopsticks) during meals was significantly associated with lower scores in the IES-R (B= −0.29, 95% CI: −0.50 to −0.09) and the DASS stress (B = −0.18, 95% CI: −0.31 to −0.06), anxiety (B = −0.36, 95% CI: −0.55 to −0.17), and depression subscales (B = −0.31, 95% CI: −0.46 to −0.15). Similarly, washing hands immediately after coughing, sneezing, or rubbing the nose was significantly associated with lower scores in the IES-R (B = −0.47, 95% CI: −0.77 to −0.17) and the DASS stress (B = −0.31, 95% CI: −0.49 to −0.13), anxiety (B = −0.63, 95 CI: −0.91 to −0.35), and depression subscales (B = −0.38, 95% CI: −0.6 to −0.16). Washing hands with soap was significantly associated with lower scores in the DASS stress (B =−0.34, 95% CI: −0.60 to −0.09), anxiety (B = −0.54, 95% CI: −0.94 to −0.14), and depression subscales (B = −0.39, 95% CI: −0.71 to −0.07). Infrequency of wearing masks regardless of the presence or absence of symptoms was significantly associated with higher IES-R scores (B = 0.52, 95% CI: 0.04 to 1.01). High frequency of wearing masks regardless of the presence or absence of symptoms was significantly associated with lower scores in the DASS anxiety (B = −0.43, 95% CI: −0.81 to −0.06) and depression subscales (B = −0.37, 95% CI: −0.67 to −0.07). Washing hands after touching contaminated objects was significantly associated with lower DASS depression scores (B = −0.53, 95% CI: −0.96 to −0.10). The majority of respondents stayed at home for 20–24 h per day (84.7%) to avoid COVID-19.

### 3.10. Additional Health Information Required and Psychological Impact

Table 6 shows additional health information required by respondents. Nearly all respondents desired additional information about COVID-19, most frequently with respect to the route of transmission (96.9%), the availability and effectiveness of medicines/vaccines (96.8%), travel advice (95.9%), overseas experience in handling COVID-19 (94.5%), the number of infected cases and locations (94.1%), advice on prevention of the COVID-19(93.7%), more tailored information (e.g., for people with chronic illnesses) (93.6%), outbreaks in the local area (92.7%), and details on symptoms of COVID-19 infection (91.6%). About 96.9% of respondents preferred regular updates for the latest information and these were found to be significantly associated with lower DASS anxiety subscale scores (B = −0.62, 95% CI: −1.00 to −0.24). Additional information on the availability and effectiveness of medicines/vaccines (B = −0.63, 95% CI: −0.99 to −0.26), the number of infections and locations (B = −0.30, 95% CI: −0.57 to −0.02), and the routes of transmission (B = −0.39, 95% CI: −0.77 to −0.02) were significantly associated with lower scores in DASS anxiety subscale. Additional information on availability and effectiveness of medicines/vaccines was significantly associated with lower scores in the DASS depression subscale (B = −0.35, 95% CI: −0.65 to −0.06).

## 4. Discussion

Our findings suggest that with respect to the initial psychological responses of the general public from 31 January to 2 February 2020, just two weeks into the country’s outbreak of COVID-19 and one day after WHO declared public health emergency of international concern, 53.8% of respondents rated the psychological impact of outbreak as moderate or severe; 16.5% of respondents reported moderate to severe depressive symptoms; 28.8% of respondents reported moderate to severe anxiety symptoms; and 8.1% reported moderate to severe stress levels. The prevalence of moderate or severe psychological impact as measured by IES-R was higher than the prevalence of depression, anxiety, and stress as measured by the DASS-21. The difference between IES-R and DASS-21 is due to the fact that the IES-R assesses the psychological impact after an event. In this study, respondents might refer the COVID-19 outbreak as the event while the DASS-21 did not specify any such event.

In this study, the majority of respondents spent 20–24 h per day at home (84.7%), did not report any physical symptoms (60.81%), and presented with good self-rated health status (68.3%). In this study, very few respondents had a direct or indirect contact history with individuals with confirmed or suspected COVID-19, or had undergone medical consultations related to COVID-19 (≤1%). The majority of respondents (>70%) were worried about their family members contracting COVID-19, but they believed that they would survive if infected.

Overall, the Internet (93.5%) was the primary health information channel for the general public during the initial stage of COVID-19 epidemic in China. Nearly all respondents (>90%) requested regular updates on the latest information on the route of transmission, availability and effectiveness of medicines/vaccines, travel advice, overseas experience in handling COVID-19, number of cases and location, advice on prevention, more tailored information (e.g., for people with chronic illnesses), information on outbreaks in the local area, and details on symptoms. The majority of respondents (>70%) were satisfied with the amount of health information available. More than half of the respondents washed their hands with soaps after touching contaminated objects, covered their mouth when coughing or sneezing, and wore masks regardless of the presence or absence of symptoms as precaution strategies.

As the COVID-19 epidemic continues to spread, our findings will provide vital guidance for the development of a psychological support strategy and areas to prioritize in China and other places which are affected by the epidemic. As the epidemic is ongoing, it is important to prepare health care systems and the general public to be medically and psychologically ready if widespread transmission occurs outside China [32]. Our findings have clinical and policy implications. First, health authorities need to identify high-risk groups based on sociodemographic information for early psychological interventions. Our sociodemographic data suggest that females suffered a greater psychological impact of the outbreak as well as higher levels of stress, anxiety, and depression. This finding corresponds to previously extensive epidemiological studies which found that women were at higher risk of depression [33]. Students were also found to experience a psychological impact of the outbreak and higher levels of stress, anxiety, and depression. As the total number of people infected by COVID-19 currently surpasses those stricken by the 2003 SARS-CoV epidemic, major cities in China have shut down schools at all levels indefinitely. The uncertainty and potential negative impact on academic progression could have an adverse effect on the mental health of students. During the epidemic, education authorities need to develop online portals and web-based applications to deliver lectures or other teaching activities [34]. As young people are more receptive towards smartphone applications [35], health authorities could consider providing online or smartphone-based psychoeducation and psychological interventions (e.g., cognitive behavior therapy, CBT) to reduce risk of virus transmission by face-to-face therapy. Online platforms could also provide a support network for those people spending most of their time at home during the epidemic. We found that the general public with no formal education had a greater likelihood of depression during the epidemic. Local agencies need to provide information in a diagrammatic or audio format in simple languages to support those with no educational background during the epidemic.

Second, health authorities need to identify the immediate psychological needs of the general population presenting with physical symptoms during the epidemic. Our results revealed that the general population presenting with specific symptoms including chills, coryza, cough, dizziness, myalgia, and sore throat, as well as those with poor self-rated health status and history of chronic illnesses, experienced a psychological impact of the outbreak and higher levels of stress, anxiety, and depression. After presentation to the clinic or hospital with the above physical symptoms, patients may be sent home, quarantined, or admitted for further investigation. Health professionals should take the opportunity to provide resources for psychological support and interventions for those who present with the above symptoms, especially during hospitalization. Taking a family history is essential, and health professionals should enquire about the level of concern for other family members, especially children, of contracting COVID-19, as these concerns are associated with stress and anxiety, respectively.

Third, government and health authorities need to provide accurate health information during the epidemic to reduce the impact of rumors [23]. Higher satisfaction with the health information received was associated with a lower psychological impact of the outbreak and lower levels of stress, anxiety, and depression. The content of health information provided during the epidemic needs to be based on evidence to avoid adverse psychological reactions. Our results showed that up-to-date and accurate health information, especially on the number of recovered individuals, was associated with lower stress levels. Additional information on medicines or vaccines, routes of transmission, and updates on the number of infected cases and location (e.g., real-time, online tracking map) were associated with lower levels of anxiety.

Fourth, the content of psychological interventions (for example CBT) needs to be modified to suit the needs of the general population during the epidemic. CBT should preferably be delivered online or via telephone to avoid the spread of infection. As online CBT does not require the presence of mental health professionals (e.g., psychologists), this will be helpful to the general public in China as there is a shortage of psychologists. Based on our findings, cognitive therapy can provide information or evidence to enhance confidence in the doctor’s ability to diagnose COVID-19. Cognitive therapy can challenge cognitive bias when recipients overestimate the risk of contracting and dying from COVID-19. As the majority of the general population in this study was homebound for 20–24 h per day during the epidemic, behavior therapy could focus on relaxation exercises to counteract anxiety and activity scheduling (e.g., home-based exercise and entertainment) to counteract depression in the home environment. Self-administered acupressure and emotional freedom techniques derived from key principles within traditional Chinese medicine are potential interventions which may benefit the mental health of general public during the COVID-19 outbreak. Further research Is required to evaluate the effectiveness of these interventions.

Fifth, our findings suggest that the precautionary measures adopted to prevent the spread of COVID-19 could have had protective psychological effects during the early stage of the epidemic. During the 2003 SARS-CoV epidemic, researchers found that moderate levels of anxiety were associated with higher uptake of preventive measures by respondents [21]. Our findings showed the opposite trend. Specific precautionary measures including avoidance of sharing utensils (e.g., chopsticks), hand hygiene, and wearing masks regardless of the presence or absence of symptoms were associated with lower levels of psychological impact, depression, anxiety, and stress. The experiences of the 2003 SARS-CoV epidemic could have changed the perception of the general public towards precautionary measures and have led to a positive effect on the initial psychological responses to the COVID-19 epidemic by giving respondents confidence and sense of control in prevention. As the Chinese prefer to use chopsticks to pick up food commonly shared on a plate during mealtime as part of their culture, it is not unexpected that avoidance of sharing utensils (e.g., chopsticks) during meals is significantly associated with less psychological impact and lower levels of anxiety, depression, and stress. During the initial stage of the COVID-19 epidemic, health authorities outside China had different recommendations for mask usage due to a global shortage of masks. While some health authorities urged citizens not to wear masks if they were well (e.g., Singapore), other health authorities urged their citizens to always have masks and hand sanitizers ready (e.g., Malaysia, Vietnam) [36]. The official guidance from the World Health Organization (WHO) advises that healthy people should only wear masks if they are taking care of a person with suspected COVID-19 infection or if people are coughing and sneezing [37]. Our study found that wearing masks, regardless of the presence or absence of symptoms, was associated with lower levels of anxiety and depression. Although the WHO emphasizes that masks are effective only when used in combination with frequent hand-cleaning with alcohol-based hand rub or soap and water, wearing a mask regardless of the presence or absence of symptoms could offer potential psychological benefits by offering a sense of security. This finding was anticipated because wearing face masks is a common practice when people are sick or to counter urban pollution or haze in parts of Asia, including China [38]. Governments and health authorities should ensure there are infrastructures to produce and provide an adequate supply of masks, soaps, alcohol-based hand rubs, and other personal hygiene products during the COVID-19 epidemic.

This study has several limitations. Given the limited resources available and time-sensitivity of the COVID-19 outbreak, we adopted the snowball sampling strategy. The snowballing sampling strategy was not based on a random selection of the sample, and the study population did not reflect the actual pattern of the general population. Furthermore, it would be ideal to conduct a prospective study on the same group of participants after a period. Due to ethical requirements on anonymity and confidentiality, we were not allowed to collect contact details and personal information from the respondents. As a result, we could not conduct a prospective study that would provide a concrete finding to support the need for a focused public health initiative. There was an oversampling of a particular network of peers (e.g., students), leading to selection bias. As a result, the conclusion was less generalizable to the entire population, particularly less educated people. Another limitation is that self-reported levels of psychological impact, anxiety, depression and stress may not always be aligned with assessment by mental health professionals. Similarly, respondents might have provided socially desirable responses in terms of the satisfaction with the health information received and precautionary measures. Lastly, the number of respondents with contact history and who had sought medical consultations was very small. Our findings could not be generalized to confirmed or suspected cases of COVID-19. Notwithstanding the above limitations, this study provides invaluable information on the initial psychological responses 2 weeks after the outbreak of COVID-19 from respondents across 194 cities in China. Our results could be used as a historical reference. Most importantly, our findings directly inform the development of psychological interventions that can minimize psychological impact, anxiety, depression, and stress during the outbreak of COVID-19 and provide a baseline for evaluating prevention, control, and treatment efforts throughout the remainder of the COVID-19 epidemic, which is still ongoing at the time of preparing this manuscript.

## 5. Conclusions

During the initial phase of COVID-19 outbreak in China, more than half of the respondents rated their psychological impact as moderate-to-severe, and about one-third reported moderate-to-severe anxiety. Female gender, student status, and specific physical symptoms were associated with a greater psychological impact of the outbreak and higher levels of stress, anxiety, and depression. Specific up-to-date and accurate health information and certain precautionary measures were associated with a lower psychological impact of the outbreak and lower levels of stress, anxiety, and depression. Our findings can be used to formulate psychological interventions to improve mental health and psychological resilience during the COVID-19 epidemic.

## Figures and Tables

**Figure 1 ijerph-17-01729-f001:**
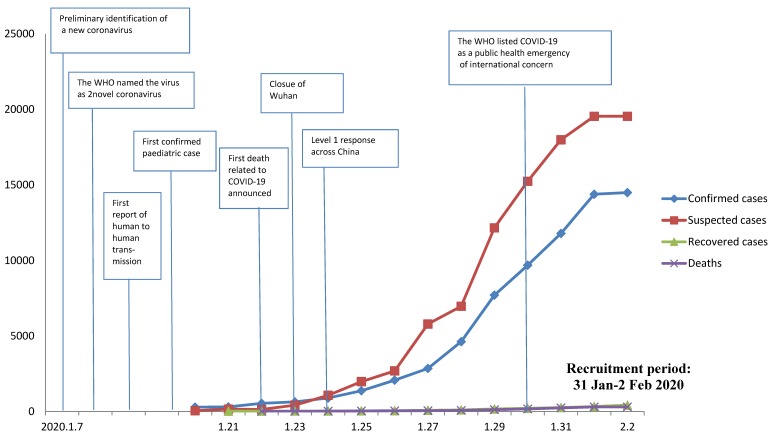
National epidemic trend of the 2019 coronavirus disease (COVID-19) outbreak in China from 7 January to 2 February 2020.

**Table 1 ijerph-17-01729-t001:** Association between demographic variables and the psychological impact of the 2019 coronavirus disease (COVID-19) outbreak as well as adverse mental health status during the epidemic (*n* = 1210).

Variables	*N* (%)	Impact of event			Stress			Anxiety			Depression		
		R-Squared (R^2^)	Adjusted R-Squared (AR^2^)	Beta (95% Confidence Interval) B (95% CI)	R^2^	AR^2^	B (95%CI)	R^2^	AR^2^	B (95% CI)	R^2^	AR^2^	B (95% CI)
**Gender**													
Male	396 (32.7)	0.005	0.005	−0.20 * (−0.35 to −0.05)	0.004	0.004	0.10 * (0.02 to 0.19)	0.006	0.005	0.19 ** (0.05 to 0.33)	0.004	0.003	0.12 * (0.01 to 0.23)
Female	814 (67.3)			Reference			Reference			Reference			Reference
**Age (Years)**													
(12–21.4)	344 (28.4)	0.009	0.006	0.21 (−0.20 to 0.62)	0.011	0.007	0.08 (−0.16 to 0.32)	0.007	0.004	0.10 (−0.28 to 0.48)	0.013	0.009	0.06 (−0.25 to 0.37)
(21.4–30.8)	643 (53.1)			0.09 (−0.31 to 0.50)			0.12 (−0.12 to 0.36)			0.07 (−0.31 to 0.44)			0.18 (−0.12 to 0.47)
(30.8–40.2)	94 (7.8)			−0.17 (−0.64 to 0.29)			−0.07 (−0.35 to 0.20)			−0.16 (−0.59 to 0.27)			−0.06 (−0.41 to 0.28)
(40.2–49.6)	90 (7.4)			−0.16 (−0.63 to 0.30)			−0.12 (−0.39 to 0.16)			−0.23 (−0.67 to 0.20)			−0.16 (−0.51 to 0.19)
(49.6–59)	39 (3.2)			Reference			Reference			Reference			Reference
**Status as a parent**													
Has a child 16 years or under	234 (19.3)	0.001	<0.001	0.04 (−0.16 to 0.25)	0.002	<0.001	−0.02 (−0.14 to 0.10)	0.003	0.001	0.08 (−0.11 to 0.27)	0.002	0.001	0.05 (−0.10 to 0.20)
Has a child older than 16 years	581 (48.1)			−0.06 (−0.22 to 0.10)			−0.07 (−0.17 to 0.02)			−0.08 (−0.23 to 0.07)			−0.06 (−0.18 to 0.06)
No children	395 (32.6)			Reference			Reference			Reference			Reference
**Marital status**													
Single	273 (22.6)	0.002	−0.001	−0.04 (−1.46 to 1.38)	0.003	0.001	0.02 (−0.81 to 0.86)	0.003	<0.001	0.71 (−0.61 to 2.03)	0.003	<0.001	0.45 (−0.60 to 1.51)
Married	925 (76.4)			0.09 (−1.33 to 1.50)			0.12 (−0.71 to 0.96)			0.80 (−0.51 to 2.12)			0.56 (−0.50 to 1.61)
Divorced/separated	9 (0.7)			0.11 (−1.52 to 1.74)			<0.001 (−0.96 to 0.96)			0.44 (−1.07 to 1.96)			0.44 (−0.77 to 1.66)
Widowed	3 (0.2)			Reference			Reference			Reference			Reference
**Household size**													
Six people or more	171 (14.1)	0.002	<0.001	0.38 (−0.39 to 1.14)	0.002	<0.001	−0.23 (−0.68 to 0.22)	<0.001	−0.002	−0.17 (−0.87 to 0.54)	0.002	<0.001	−0.19 (−0.76 to 0.38)
Three to five people	976 (80.7)			0.25 (−0.49 to 0.99)			−0.20 (−0.63 to 0.24)			−0.12 (−0.81 to 0.57)			−0.09 (−0.64 to 0.47)
Two people	52 (4.3)			0.41 (−0.40 to 1.22)			−0.33 (−0.81 to 0.15)			−0.18 (−0.93 to 0.58)			−0.21 (−0.82 to 0.39)
One person	11 (0.9)			Reference			Reference			Reference			Reference
**Employment status**													
Unemployed	67 (5.5)	0.009	0.006	0.13 (−0.19 to 0.45)	0.007	0.004	0.12 (−0.07 to 0.31)	0.007	0.003	0.21 (−0.09 to 0.51)	0.005	0.001	0.16 (−0.08 to 0.40)
Farmers	24 (2)			−0.08 (−0.59 to 0.43)			0.003 (−0.30 to 0.30)			0.07 (−0.41 to 0.54)			−0.07 (−0.45 to 0.31)
Retired	7 (0.6)			−0.76 (−1.69 to 0.17)			−0.37 (−0.92 to 0.18)			−0.54 (−1.41 to 0.32)			−0.48 (−1.18 to 0.21)
Student	639 (52.8)			0.20 * (0.05 to 0.35)			0.11 * (0.02 to 0.19)			0.16 * (0.02 to 0.30)			0.08 * (−0.03 to 0.19)
Employed	473 (39.1)			Reference			Reference			Reference			Reference
**Educational attainment**													
None	2 (0.2)	0.008	0.003	−0.07 (−1.88 to 1.74)	0.004	−0.001	0.76 (−0.30 to 1.83)	0.004	−0.001	1.02 (−0.66 to 2.71)	0.008	0.003	1.81 ** (0.46 to 3.16)
Primary school	8 (0.7)			−1.07 *(−2.09 to −0.06)			−0.11 (−0.71 to 0.49)			−0.10 (−1.05 to 0.84)			−0.07 (−0.82 to 0.69)
Lower secondary school	55 (4.5)			0.21 (−0.42 to 0.84)			0.20 (−0.17 to 0.57)			0.38 (−0.21 to 0.96)			0.41 (−0.06 to 0.88)
Upper secondary school	81 (6.7)			0.01 (−0.59 to 0.61)			0.16 (−0.20 to 0.51)			0.36 (−0.19 to 0.92)			0.34 (−0.11 to 0.79)
University—Bachelors	805 (66.5)			0.19 (−0.35 to 0.73)			0.21 (−0.11 to 0.53)			0.32 (−0.18 to 0.82)			0.35 (−0.05 to 0.75)
University—Masters	238 (19.7)			0.14 (−0.42 to 0.69)			0.18 (−0.15 to 0.51)			0.24 (−0.28 to 0.76)			0.33 (−0.09 to 0.74)
University—Doctorate	21 (1.7)			Reference			Reference			Reference			Reference

* *p* < 0.05; ** *p* < 0.01.

**Table 2 ijerph-17-01729-t002:** Association between physical health status in the past 14 days and the psychological impact of the 2019 coronavirus disease (COVID-19) outbreak as well as adverse mental health status during the epidemic (*n* = 1210).

Variable	*n* (%)	Impact of Event			Stress			Anxiety			Depression		
		R^2^	AR^2^	B (95% CI)	R^2^	AR^2^	B (95% CI)	R^2^	A^R2^	B (95% CI)	R^2^	A^R2^	B (95% CI)
**Persistent fever (>38°C for at least 1 day)**													
Yes	6 (0.5)	<0.001	−0.001	−0.23 (−1.23 to 0.78)	0.001	0.001	0.40 (−0.19 to 0.99)	0.006	0.005	1.23 * (0.30 to 2.15)	0.005	0.005	0.98 * (0.23 to 1.72)
No	1204 (99.5)			Reference			Reference			Reference			Reference
**Chills**													
Yes	42 (3.5)	0.005	0.004	0.46 * (0.08 to 0.84)	0.012	0.011	0.44 *** (0.22 to 0.67)	0.009	0.008	0.60 **(0.24 to 0.96)	0.007	0.006	0.41 **(0.13 to 0.70)
No	1168 (96.5)			Reference			Reference			Reference			Reference
**Headache**													
Yes	117 (9.7)	0.008	0.007	0.37 ** (0.13 to 0.61)	0.002	0.001	0.12 (−0.02 to 0.26)	0.008	0.008	0.36 ** (0.14 to 0.58)	0.005	0.004	0.23 * (0.05 to 0.40)
No	1093 (90.3)			Reference			Reference			Reference			Reference
**Myalgia**													
Yes	95 (7.9)	0.019	0.018	0.63 ***(0.37 to 0.89)	0.025	0.024	0.43 ***(0.28 to 0.59)	0.025	0.025	0.69 *** (0.45 to 0.93)	0.021	0.02	0.50 *** (0.31 to 0.69)
No	1115(92.1)			Reference			Reference			Reference			Reference
**Cough**													
Yes	168 (13.9)	0.009	0.008	0.33 ** (0.13 to 0.54)	0.008	0.007	0.19 ** (0.07 to 0.31)	0.007	0.006	0.29 ** (0.10 to 0.47)	0.006	0.005	0.21 **(0.06 to 0.36)
No	1042 (86.1)			Reference			Reference			Reference			Reference
**Breathing difficulty**													
Yes	5 (0.4)	0.002	0.001	0.88 (−0.22 to 1.97)	0.002	0.002	0.57 (−0.07 to 1.22)	0.008	0.007	1.63 ** (0.61 to 2.64)	0.008	0.007	1.28 ** (0.46 to 2.09)
No	1205 (99.6)			Reference			Reference			Reference			Reference
**Dizziness**													
Yes	88 (7.3)	0.013	0.012	0.54 *** (0.27 to 0.81)	0.014	0.013	0.33 *** (0.17 to 0.49)	0.020	0.019	0.63 *** (0.38 to 0.88)	0.014	0.013	0.42 *** (0.22 to 0.62)
No	1122 (92.7)			Reference			Reference			Reference			Reference
**Coryza**													
Yes	205 (16.9)	0.014	0.013	0.39 *** (0.20 to 0.58)	0.016	0.015	0.25 *** (0.14 to 0.36)	0.022	0.021	0.46 ***(0.28 to 0.63)	0.018	0.017	0.33 *** (0.19 to 0.47)
No	1005 (83.1)			Reference			Reference			Reference			Reference
**Sore throat**													
Yes	139 (11.5)	0.007	0.007	0.34 ** (0.12 to 0.56)	0.005	0.004	0.16 *(0.03 to 0.29)	0.009	0.008	0.35 ** (0.14 to 0.55)	0.004	0.003	0.17 * (0.01 to 0.34)
No	1071 (88.5)			Reference			Reference			Reference			Reference
**Persistent fever and cough or difficulty breathing**													
Yes	4 (0.3)	<0.001	−0.001	−0.23 (−1.45 to 1.00)	0.001	<0.001	0.32 (−0.40 to 1.04)	0.002	0.002	0.98 (−0.16 to 2.11)	<0.001	−0.001	0.22 (−0.69 to 1.13)
No	1206 (99.7)			Reference			Reference			Reference			Reference
**Consultation with doctor in the clinic in the past 14 days**													
Yes	42 (3.5)	<0.001	−0.001	−0.06 (−0.44 to 0.32)	0.002	0.001	0.17 (−0.06 to 0.40)	0.004	0.003	0.38 * (0.02 to 0.73)	0.002	0.001	0.22 (−0.07 to 0.50)
No	1168 (96.5)			Reference			Reference			Reference			Reference
**Recent hospitalization in the past 14 days**													
Yes	4 (0.3)	0.001	<0.001	0.78 (−0.45 to 2.00)	0.001	<0.001	0.32 (−0.40 to 1.04)	0.004	0.003	1.23 * (0.09 to 2.36)	<0.001	−0.001	−0.28 (−1.19 to 0.63)
No	1206 (99.7)			Reference			Reference			Reference			Reference
**Recent testing for COVID-19 in the past 14 days**													
Yes	11 (0.9)	<0.001	−0.001	−0.18 (−0.92 to 0.56)	<0.001	−0.001	−0.07 (−0.51 to 0.37)	<0.001	<0.001	0.22 (−0.47 to 0.91)	<0.001	−0.001	0.02 (−0.54 to 0.57)
No	1199 (99.1)			Reference			Reference			Reference			Reference
**Recent quarantine in the past 14 days**													
Yes	26 (2.1)	0.001	0.001	0.32 (−0.16 to 0.81)	<0.001	−0.001	−0.01 (−0.30 to 0.28)	<0.001	−0.001	0.03 (−0.42 to 0.48)	<0.001	−0.001	−0.11 (−0.47 to 0.25)
No	1184 (97.9)			Reference			Reference			Reference			Reference
**Current self-rating health status**													
Poor/Very poor	11 (1)	0.021	0.018	0.76 * (0.02 to 1.49)	0.034	0.032	0.45 * (0.02 to 0.88)	0.034	0.032	0.90 * (0.22 to 1.58)	0.030	0.027	0.65 * (0.1 to 1.20)
Average	372 (30.8)			0.37 *** (0.21 to 0.52)			0.19 *** (0.11 to 0.28)			0.41 *** (0.27 to 0.55)			0.26 *** (0.15 to 0.38)
Good/Very good	827 (68.3)			Reference			Reference			Reference			Reference
**Chronic illness**													
Yes	78 (6.4)	0.003	0.003	0.29 * (0.01 to 0.58)	0.006	0.005	0.24 ** (0.07 to 0.41)	0.011	0.010	0.48 *** (0.22 to 0.75)	0.010	0.009	0.38 *** (0.17 to 0.59)
No	1132 (93.6)			Reference			Reference			Reference			Reference
**Medical insurance coverage**													
Yes	1118 (92.4)	<0.001	<0.001	0.09 (−0.18 to 0.36)	<0.001	−0.001	−0.003 (−0.16 to 0.15)	<0.001	−0.001	−0.04 (−0.29 to 0.21)	<0.001	−0.001	0.02 (−0.18 to 0.22)
No	92 (7.6)			Reference			Reference			Reference			Reference

* *p* < 0.05; ** *p* < 0.01; *** *p* < 0.001.

**Table 3 ijerph-17-01729-t003:** Association between contact history in the past 14 days and the psychological impact of the 2019 coronavirus disease (COVID-19) outbreak as well as adverse mental health status during the epidemic (*n* = 1210).

Variables	*n* (%)	Impact of Event			Stress			Anxiety			Depression		
		R^2^	AR^2^	B (95% CI)	R^2^	AR^2^	B (95% CI)	R^2^	AR^2^	B (95% CI)	R^2^	AR^2^	B (95% CI)
**Close contact with an individual with confirmed infection with COVID-19**													
Yes	4 (0.3)	0.001	<0.001	0.53 (−0.70 to 1.75)	0.001	<0.001	0.32 (−0.40 to 1.04)	0.002	0.002	0.98 (−0.16 to 2.11)	0.004	0.003	0.97 * (0.06 to 1.88)
No	1206 (99.7)			Reference			Reference			Reference			Reference
Indirect contact with an individual with confirmed infection with COVID-19													
Yes	6 (0.5)	<0.001	−0.001	−0.06 (−1.06 to 0.94)	0.001	<0.001	−0.27 (−0.86 to 0.32)	<0.001	−0.001	−0.28 (−1.21 to 0.65)	0.001	<0.001	−0.37 (−1.11 to 0.38)
No	1204 (99.5)			Reference			Reference			Reference			Reference
Contact with an individual with suspected COVID-19 or infected materials													
Yes	12 (1.0)	0.001	<0.001	0.36 (−0.35 to 1.07)	0.003	0.002	0.41 (−0.01 to 0.82)	0.007	0.006	0.98 ** (0.32 to 1.64)	0.008	0.007	0.81 ** (0.29 to 1.34)
No	1198 (99.0)			Reference			Reference			Reference			Reference
											<0.001	−0.001	−0.03 (−1.32 to 1.26)
													Reference

* *p* < 0.05, ** *p* < 0.01.

**Table 4 ijerph-17-01729-t004:** Association between knowledge and concerns about the 2019 coronavirus disease (COVID-19) and the psychological impact of outbreak as well as adverse mental health status during the epidemic (*n* = 1210).

Variables	*n* (%)	Impact of Event			Stress				Anxiety				Depression				
		R^2^	AR^2^	B (95% CI)	R^2^	AR^2^	B (95% CI)		R^2^	AR^2^	B (95% CI)		R^2^	AR^2^	B (95% CI)		
**Knowledge of COVID-19**																	
**Route of transmission**																	
Droplets																	
Agree	1115 (92.1)	0.002	0.001	0.21 (−0.07 to 0.49)	0.003	0.001	0.15 (−0.01 to 0.32)		0.001	<0.001	0.17 (−0.09 to 0.43)		0.005	0.004	0.27 * (0.06 to 0.48)		
Disagree	13 (1.1)			0.48 (−0.25 to 1.21)			0.09 (−0.34 to 0.52)				0.22 (−0.45 to 0.90)				0.18 (−0.36 to0.72)		
Do not know	82 (6.8)			Reference			Reference				Reference				Reference		
**Contact via contaminated objects**																	
Agree	892 (73.7)	<0.001	−0.001	0.04 (−0.15 to 0.22)	0.003	0.001	−0.02 (−0.13 to 0.09)		0.002	0.001	−0.07 (−0.24 to 0.10)		0.001	−0.001	0.02 (−0.12 to 0.15)		
Disagree	94 (7.8)			−0.04 (−0.34 to 0.26)			−0.16 (−0.34 to 0.02)				−0.23 (−0.51 to 0.05)				−0.10 (−0.33 to 0.12)		
Do not know	224 (18.5)			Reference			Reference				Reference				Reference		
Airborne																	
Agree	732 (60.5)	0.002	<0.001	0.11 (−0.07 to 0.29)	0.001	−0.001	0.04 (−0.07 to 0.14)		0.002	<0.001	0.12 (−0.05 to 0.28)		0.001	<0.001	0.08 (−0.05 to 0.22)		
Disagree	225 (18.6)			0.17 (−0.05 to 0.40)			−0.002 (−0.13 to 0.13)				0.04 (−0.17 to 0.25)				0.03 (−0.14 to 0.20)		
Do not know	253 (20.9)			Reference			Reference				Reference				Reference		
**Have you heard that the number of infected COVID-19 individuals has increased?**																	
Heard	1195 (98.8)	0.001	<0.001	0.40 (−0.24 to 1.03)	<0.001	−0.001	0.10 (−0.27 to 0.47)		<0.001	−0.001	−0.16 (−0.75 to 0.43)		<0.001	−0.001	−0.14 (−0.61 to 0.34)		
Not heard	15 (1.2)			Reference			Reference				Reference				Reference		
**Have you heard that the number of COVID-19 deaths has increased?**																	
Heard	1183 (97.8)	0.001	<0.001	0.21 (−0.27 to 0.69)	0.001	<0.001	0.18 (−0.10 to 0.46)		<0.001	−0.001	0.04 (−0.40 to 0.48)		<0.001	−0.001	0.09 (−0.27 to 0.44)		
Not heard	27 (2.2)			Reference			Reference				Reference				Reference		
**Have you heard that the number of individuals that have recovered from COVID-19 infection has increased?**																	
Heard	1129 (93.3)	0.001	0.001	−0.19 (−0.47 to 0.09)	0.007	0.006	−0.24 ** (−0.40 to −0.07)		0.003	0.002	−0.25 (−0.51 to 0.01)		0.004	0.003	−0.24 * (−0.45 to −0.03)		
Not heard	81 (6.7)			Reference			Reference				Reference				Reference		
**The main source of health information**																	
Internet	1131 (93.5)	0.003	<0.001	−0.46 (−1.46 to 0.54)	0.007	0.004	−0.25 (−0.83 to0.34 )		0.010	0.006	−0.57 (−1.50 to 0.35)		0.007	0.004	0.19 (−0.55 to 0.94)		
Television	62 (5.1)			−0.22 (−1.26 to 0.83)			−0.07 (−0.68 to 0.54)				−0.35 (−1.32 to 0.62)				0.31 (−0.47 to 1.09)		
Radio	1 (0.1)			0.83 (−1.81 to 3.47)			1.33 (−0.22 to 2.89)				2.67 * (0.22 to 5.11)				2.67 ** (0.70 to 4.63)		
Family members	10 (0.8)			−0.47 (−1.73 to 0.80)			−0.27 (−1.01 to 0.48)				−0.33 (−1.50 to 0.84)				−0.03(−0.97 to 0.91)		
Other sources	6 (0.5)			Reference			Reference				Reference				Reference		
**Satisfaction with the amount of health information available about COVID-19**																	
Very satisfied	485 (40.1)	0.018	0.014	0.02 (−0.34 to 0.37)	0.014	0.011	−0.09 (−0.30 to 0.13)		0.013	0.010	−0.20 (−0.53 to 0.14)		0.014	0.011	−0.12 (−0.38 to 0.15)		
Somewhatsatisfied	423 (35.0)			0.23 (−0.13 to 0.59)			0.03 (−0.19 to 0.24)				−0.02 (−0.36 to 0.31)				−0.001 (−0.27 to 0.27)		
Not very satisfied	211 (17.4)			0.39 * (0.01 to 0.77)			0.09 (−0.14 to 0.31)				0.05 (−0.31 to 0.40)				0.08 (−0.21 to 0.36)		
Not satisfied at all	40 (3.3)			0.63 * (0.11 to 1.14)			0.32 * (0.02 to 0.62)				0.41 (−0.07 to 0.88)				0.43 * (0.04 to 0.81)		
Do not know	51 (4.2)			Reference			Reference				Reference				Reference		
**Concerns about COVID-19**																	
**Level of confidence in own doctor’s ability to diagnose or recognize**																	
Very confident	563 (46.5)	0.025	0.022	−0.2 (−0.66 to 0.27)	0.021	0.017	0.05 (−0.23 to 0.32)		0.024	0.021	0.02 (−0.42 to 0.45)		0.021	0.018	0.02 (−0.33 to 0.37)		
Somewhat confident	561 (46.4)			0.19 (−0.28 to 0.66)			0.16 (−0.12 to 0.44)				0.22 (−0.22 to 0.65)				0.09 (−0.26 to 0.44)		
Not very confident	50 (4.1)			0.19 (−0.39 to 0.76)			0.18 (−0.16 to 0.52)				0.38 (−0.15 to 0.91)				0.10 (−0.33 to 0.52)		
Not at all confident	8 (0.7)			0.66 (−0.31 to 1.63)			1.18 *** (0.61 to 1.75)				1.86 *** (0.96 to 2.76)				1.66 ***(0.94 to 2.38)		
Do not know	28 (2.3)			Reference			Reference				Reference				Reference		
**Likelihood of contracting COVID−19 during the current outbreak**																	
Very likely	135 (11.2)	0.019	0.016	−0.33 * (−0.61 to −0.05)	0.008	0.005	0.05 (−0.11 to 0.22)		0.009	0.005	0.07 (−0.20 to 0.33)		0.007	0.004	0.15 (−0.06 to 0.36)		
Somewhat likely	358 (29.6)			0.15 (−0.09 to 0.38)			0.06 (−0.08 to 0.20)				−0.02 (−0.23 to 0.20)				0.04 (−0.14 to 0.21)		
Not very likely	437 (36.1)			0.14 (−0.09 to 0.36)			−0.002 (−0.14 to 0.13)				−0.05 (−0.26 to 0.16)				0.03 (−0.14 to 0.20)		
Not likely at all	121 (10.0)			−0.23(−0.52 to 0.06)			−0.18 * (−0.35 to −0.01)				−0.36 * (−0.63to−0.09)				−0.19 (−0.41 to 0.03)		
Do not know	159 (13.1)			Reference			Reference				Reference				Reference		
**Likelihood of surviving if infected with COVID-19**																	
Very likely	278 (23.0)	0.014	0.011	−0.19 (−0.41 to 0.03)	0.009	0.006	−0.02 (−0.15 to 0.11)		0.006	0.002	−0.06 (−0.27 to 0.14)		0.007	0.003	0.01 (−0.15 to 0.17)		
Somewhat likely	559 (46.2)			0.12 (−0.07 to 0.31)			0.01 (−0.10 to 0.13)				−0.03 (−0.21 to 0.15)				−0.01 (−0.15 to 0.14)		
Not very likely	124 (10.2)			0.23 (−0.04 to 0.50)			0.18 * (0.02 to 0.34)				0.18 (−0.08 to 0.43)				0.15 (−0.06 to 0.35)		
Not likely at all	20 (1.7)			0.42 (−0.15 to 0.99)			0.34 * (0.01 to 0.68)				0.42 (−0.11 to 0.95)				0.49 * (0.07 to 0.92)		
Do not know	229 (18.9)			Reference			Reference				Reference				Reference		
**Concerns about other family members getting COVID−19 infection**																	
Very worried	417 (34.5)	0.017	0.014	0.75(−0.03 to 1.53)	0.007	0.004	0.50^*^(0.04 to 0.96)		0.006	0.003	0.59(−0.13 to 1.32)		0.005	0.001	0.29(−0.30 to 0.87)		
Somewhat worried	492 (40.7)			0.67 (−0.10 to 1.45)			0.40 (−0.05 to 0.86)				0.43 (−0.30 to 1.15)				0.20 (−0.38 to 0.78)		
Not very worried	221 (18.3)			0.44 (−0.34 to 1.23)			0.43 (−0.04 to 0.89)				0.44 (−0.30 to 1.17)				0.26 (−0.33 to 0.85)		
Not worried at all	70 (5.8)			0.19 (−0.64 to 1.01)			0.33 (−0.16 to 0.81)				0.36 (−0.41 to 1.13)				0.04 (−0.57 to 0.66)		
Do not have family members	10 (0.8)			Reference			Reference				Reference				Reference		
**Concerns about a child younger than 16 years getting COVID-19 infection**																	
Very worried	309 (25.5)	0.006	0.003	0.25 * (0.05 to 0.44)	0.001	−0.003	0.05 (−0.07 to 0.16)		0.007	0.004	0.24 ** (0.07 to 0.42)		0.002	−0.001	0.09 (−0.05 to 0.24)		
Somewhat worried	307 (25.4)			0.13 (−0.06 to 0.32)			0.03 (−0.09 to 0.14)				0.21 * (0.03 to 0.39)				0.08 (−0.06 to 0.23)		
Not very worried	151 (12.5)			0.10 (−0.14 to 0.34)			0.04 (−0.10 to 0.18)				0.21 (−0.01 to 0.43)				0.08 (−0.09 to 0.26)		
Not worried at all	102 (8.4)			−0.02 (−0.30 to 0.26)			0.004 (−0.16 to 0.17)				0.14 (−0.12 to 0.40)				0.03 (−0.18 to 0.23)		
Do not have children	341 (28.2)			Reference			Reference				Reference				Reference		

* *p* < 0.05; ** *p* < 0.01; *** *p* < 0.001.

**Table 5 ijerph-17-01729-t005:** Association between precautionary measures in the past 14 days and the psychological impact of the 2019 coronavirus disease (COVID-19) outbreak as well as adverse mental health status during the epidemic (*n* = 1210).

Variables	*n* (%)	Impact of Event			Stress			Anxiety			Depression		
		R^2^	AR^2^	B (95% CI)	R^2^	AR^2^	B (95% CI)	R^2^	AR^2^	B (95% CI)	R^2^	AR^2^	B (95% CI)
**Covering mouth when coughing and sneezing**													
Always	694 (57.4)	0.009	0.006	0.02 (−0.34 to 0.37)	0.003	<0.001	0.02 (−0.19 to 0.23)	0.007	0.004	−0.19 (−0.52 to 0.14)	0.001	−0.002	−0.09 (−0.35 to 0.18)
Most of the time	282 (23.3)			0.18 (−0.19 to 0.55)			0.09 (−0.13 to 0.31)			−0.09 (−0.43 to 0.26)			−0.04 (−0.32 to 0.24)
Sometime	106 (8.8)			0.40 (−0.02 to 0.82)			0.12 (−0.13 to 0.36)			0.09 (−0.30 to 0.47)			0.02 (−0.30 to 0.33)
Occasionally	77 (6.3)			0.18 (−0.26 to 0.62)			−0.03 (−0.29 to 0.23)			−0.32 (−0.73 to 0.09)			−0.02 (−0.35 to 0.31)
Never	51 (4.2)			Reference			Reference			Reference			Reference
**Avoiding sharing of utensils (e.g., chopsticks) during meals**													
Always	490 (40.5)	0.041	0.037	−0.29 ** (−0.50 to −0.09)	0.017	0.013	−0.18 ** (−0.31 to −0.06)	0.017	0.013	−0.36 *** (−0.55 to −0.17)	0.016	0.013	−0.31 *** (−0.46 to −0.15)
Most of the time	207 (17.1)			0.17 (−0.07 to 0.40)			0.01 (−0.13 to 0.16)			−0.03 (−0.26 to 0.19)			−0.07 (−0.25 to 0.11)
Sometime	162 (13.4)			0.23 (−0.02 to 0.49)			−0.02 (−0.17 to 0.14)			−0.13 (−0.37 to 0.11)			−0.20 * (−0.39 to −0.003)
Occasionally	156 (12.9)			0.36 ** (0.10 to 0.62)			0.03 (−0.12 to 0.18)			−0.14 (−0.38 to 0.11)			−0.12 (−0.32 to 0.07)
Never	195 (16.1)			Reference			Reference			Reference			Reference
**Washing hands with soap and water**													
Always	684 (56.5)	0.029	0.026	−0.42 (−0.85 to 0.01)	0.011	0.007	−0.34 ** (−0.60 to −0.09)	0.015	0.012	−0.54 ** (−0.94 to −0.14)	0.011	0.007	−0.39 * (−0.71 to −0.07)
Most of the time	266 (22)			−0.12 (−0.56 to 0.33)			−0.29 * (−0.56 to −0.03)			−0.40 (−0.81 to 0.02)			−0.27 (−0.60 to 0.07)
Sometime	127 (10.5)			0.07 (−0.40 to 0.54)			−0.22 (−0.50 to 0.07)			−0.23 (−0.67 to 0.21)			−0.25 (−0.61 to 0.10)
Occasionally	100 (8.3)			0.13 (−0.35 to 0.62)			−0.17 (−0.46 to 0.12)			−0.21 (−0.67 to 0.24)			−0.15 (−0.51 to 0.22)
Never	33 (2.7)			Reference			Reference			Reference			Reference
**Washing hands immediately after coughing, rubbing nose, or sneezing**													
Always	496 (41)	0.042	0.039	−0.47 ** (−0.77 to −0.17)	0.02	0.016	−0.31 ** (−0.49 to −0.13)	0.22	0.019	−0.63 *** (−0.91 to −0.35)	0.021	0.018	−0.38 ** (−0.60 to −0.16)
Most of the time	227 (18.8)			−0.003 (−0.32 to 0.32)			−0.17 (−0.36 to 0.02)			−0.44 ** (−0.74 to −0.14)			−0.26 * (−0.50 to −0.02)
Sometime	227 (18.8)			0.02 (−0.30 to 0.34)			−0.12 (−0.32 to 0.07)			−0.41 ** (−0.71 to −0.11)			−0.18 (−0.42 to 0.06)
Occasionally	185 (15.2)			0.14 (−0.19 to 0.47)			−0.08 (−0.28 to 0.12)			−0.29 (−0.60 to 0.02)			−0.04 (−0.29 to 0.20)
Never	75 (6.2)			Reference			Reference			Reference			Reference
**Wearing mask regardless of the presence or absence of symptoms**													
Always	723 (59.8)	0.026	0.023	−0.19 (−0.59 to 0.21)	0.009	0.006	−0.21 (−0.45 to 0.02)	0.01	0.006	−0.43 * (−0.81 to −0.06)	0.015	0.012	−0.37 * (−0.67 to −0.07)
Most of the time	263 (21.7)			0.12 (−0.30 to 0.53)			−0.09 (−0.34 to 0.16)			−0.27 (−0.66 to 0.12)			−0.21 (−0.52 to 0.10)
Sometime	116 (9.6)			0.16 (−0.29 to 0.61)			−0.08 (−0.35 to 0.19)			−0.25 (−0.67 to 0.17)			−0.25 (−0.59 to 0.08)
Occasionally	69 (5.7)			0.52 * (0.04 to 1.01)			−0.04 (−0.33 to 0.24)			−0.14 (−0.60 to 0.31)			0.006 (−0.36 to 0.37)
Never	39(3.2)			Reference			Reference			Reference			Reference
**Washing hands after touching contaminated objects**													
Always	806 (66.6)	0.018	0.014	−0.11 (−0.69 to 0.47)	0.007	0.003	−0.21 (−0.56 to 0.13)	0.014	0.011	−0.52 (−1.06 to 0.02)	0.012	0.008	−0.53 * (−0.96 to −0.10)
Most of the time	283 (23.4)			0.19 (−0.40 to 0.78)			−0.15 (−0.49 to 0.21)			−0.37 (−0.92 to 0.18)			−0.41 (−0.85 to 0.03)
Sometime	66 (5.4)			0.40 (−0.25 to 1.04)			−0.01 (−0.39 to 0.38)			−0.03 (−0.63 to 0.58)			−0.27 (−0.76 to 0.21)
Occasional	37 (3.1)			0.31 (−0.39 to 1.00)			−0.07 (−0.48 to 0.34)			−0.22 (−0.87 to 0.43)			0.24 (−0.76 to 0.28)
Never	18 (1.5)			Reference			Reference			Reference			Reference
**Feeling that too much unnecessary worry has been made about the COVID-19 outbreak**													
Always	156 (12.9)	0.019	0.016	−0.47 *** (−0.69 to −0.25)	0.002	−0.001	−0.08 (−0.21 to 0.05)	0.003	<0.001	0.12 (−0.09 to 0.33)	0.005	0.002	0.12 (−0.04 to 0.29)
Most of the time	108 (8.9)			−0.19 (−0.44 to 0.07)			−0.05 (−0.20 to 0.11)			0.20 (−0.04 to 0.44)			0.20 * (0.003 to 0.39)
Sometime	242 (20)			−0.03 (−0.21 to 0.16)			−0.01 (−0.12 to 0.10)			0.07 (−0.10 to 0.25)			0.01 (−0.13 to 0.15)
Occasionally	166 (13.7)			0.13 (−0.09 to 0.34)			0.03 (−0.10 to 0.16)			0.12 (−0.08 to 0.33)			0.10 (−0.07 to 0.26)
Never	538 (44.5)			Reference			Reference			Reference			Reference
**Average number of hours staying at home per day to avoid COVID−19**													
[0–9]	39 (3.2)	0.001	<0.001	−0.15 (−0.55 to 0.25)	0.002	<0.001	−0.16 (−0.39 to 0.08)	0.001	0.001	−0.15 (−0.52 to 0.22)	0.002	0.001	−0.21 (−0.51 to 0.08)
[10–19]	146 (12.1)			0.11 (−0.10 to 0.33)			−0.03 (−0.16 to 0.10)			−0.06 (−0.26 to 0.14)			−0.08 (−0.24 to 0.08)
[20–24]	1025 (84.7)			Reference			Reference			Reference			Reference

* *p* < 0.05; ** *p* < 0.01; *** *p* < 0.001.

**Table 6 ijerph-17-01729-t006:** Association between additional health information required by participants and the psychological impact of the 2019 coronavirus disease (COVID-19) outbreak as well as adverse mental health status during the epidemic (*n* = 1210).

Variables	*n* (%)	Impact of Event			Stress			Anxiety			Depression		
		R^2^	AR^2^	B (95% CI)	R^2^	AR^2^	B (95% CI)	R^2^	AR^2^	B (95% CI)	R^2^	AR^2^	B (95% CI)
**Need for further health information about the COVID-19 infection**													
Yes	1048 (86.6)	0.010	0.009	0.36 ** (0.15 to 0.57)	0.003	0.002	0.12 * (0.00 to 0.24)	0.002	0.001	0.16 (−0.03 to 0.35)	0.001	<0.001	0. 1 (−0.06 to 0.25)
No	162 (13.4)			Reference			Reference			Reference			Reference
**Need for details on symptoms of the COVID−19 infection**													
Yes	1108 (91.6)	0.006	0.005	0.34 ** (0.09 to 0.59)	<0.001	−0.001	−0.02 (−0.17 to 0.13)	0.001	<0.001	−0.12 (−0.36 to 0.11)	0.001	<0.001	−0.11 (−0.30 to 0.08)
No	102 (8.4)			Reference			Reference			Reference			Reference
**Need for advice on prevention of the COVID−19 infection**													
Yes	1134 (93.7)	0.010	0.009	0.52 *** (0.23 to 0.81)	0.001	0.001	0.11 (−0.06 to 0.28)	0.001	<0.001	0.13 (−0.14 to 0.40)	<0.001	−0.001	0.05 (−0.17 to 0.26)
No	76 (6.3)			Reference			Reference			Reference			Reference
**Need for advice on treatment of the COVID−19 infection**													
Yes	1000 (82.6)	0.003	0.003	0.19 * (0.006 to 0.38)	<0.001	−0.001	0.03 (−0.08 to 0.14)	<0.001	−0.001	0.03 (−0.14 to 0.20)	<0.001	<0.001	−0.05 (−0.18 to 0.09)
No	210 (17.4)			Reference			Reference			Reference			Reference
**Need for regular updates for latest information about the COVID−19 infection**													
Yes	1173 (96.9)	<0.001	−0.001	−0.03 (−0.44 to 0.38)	0.001	<0.001	−0.11 (−0.35 to 0.13)	0.008	0.008	−0.62 ** (−1.00 to −0.24)	0.003	0.002	−0.29 (−0.59 to 0.01)
No	37 (3.1)			Reference			Reference			Reference			Reference
**Need for the latest updates for outbreaks of the COVID−19 infection in the local area**													
Yes	1122 (92.7)	<0.001	−0.001	0.06 (−0.21 to 0.33)	<0.001	−0.001	0.01 (−0.15 to 0.17)	0.001	<0.001	−0.10 (−0.36 to 0.15)	0.001	<0.001	0.09 (−0.11 to 0.30)
No	88 (7.3)			Reference			Reference			Reference			Reference
**Need for advice for people who may need more tailored information, such as those with pre-existing illness**													
Yes	1133 (93.6)	0.001	0.001	0.19 (−0.10 to 0.48)	<0.001	−0.001	0.004 (−0.17 to 0.17)	0.001	<0.001	−0.14 (−0.41 to 0.13)	<0.001	−0.001	−0.02 (−0.23 to 0.20)
No	77 (6.4)			Reference			Reference			Reference			Reference
**Need for information on the availability and effectiveness of medicines/vaccines for the COVID−19 infection**													
Yes	1171 (96.8)	0.001	<0.001	0.19 (−0.21 to 0.59)	0.002	0.001	−0.16 (−0.40 to 0.07)	0.009	0.008	−0.63 ** (−0.99 to −0.26)	0.005	0.004	−0.35 * (−0.65 to −0.06)
No	39 (3.2)			Reference			Reference			Reference			Reference
**Need for the latest updates on the number of people infected by COVID-19 and their location**													
Yes	1139 (94.1)	0.001	<0.001	0.17 (−0.13 to 0.47)	0.001	<0.001	−0.09 (−0.27 to 0.08)	0.004	0.003	−0.30 * (−0.57 to −0.02)	0.001	<0.001	−0.13 (−0.35 to 0.10)
No	71 (5.9)			Reference			Reference			Reference			Reference
**Need for travel advice for the COVID-19 epidemic**													
Yes	1160 (95.9)	<0.001	−0.001	0.07 (−0.29 to 0.42)	<0.001	<0.001	−0.07 (−0.28 to 0.14)	0.001	<0.001	−0.19 (−0.52 to 0.14)	<0.001	−0.001	−0.07 (−0.34 to 0.19)
No	50 (4.1)			Reference			Reference			Reference			Reference
**Need for updates on the routes of transmission of COVID-19**													
Yes	1172 (96.9)	<0.001	<0.001	0.15 (−0.25 to 0.55)	0.001	<0.001	−0.10 (−0.33 to 0.14)	0.003	0.003	−0.39 * (−0.77 to −0.02)	0.002	0.001	−0.21 (−0.51 to 0.09)
No	38 (3.1)			Reference			Reference			Reference			Reference
**Need for updates on how other countries handle the COVID-19 outbreak**													
Yes	1144 (94.5)	0.002	0.001	0.25 (−0.06 to 0.56)	<0.001	−0.001	−0.008 (−0.19 to 0.18)	0.001	<0.001	−0.14 (−0.43 to 0.15)	<0.001	<0.001	−0.08 (−0.31 to 0.15)
No	66 (5.5)			Reference			Reference			Reference			Reference

* *p* < 0.05; ** *p* < 0.01; *** *p* < 0.001.

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
