# Peer review of "Immediate Psychological Responses and Associated Factors during the Initial Stage of the 2019 Coronavirus Disease (COVID-19) Epidemic among the General Population in China"

_ijerph, 2020, doi:10.3390/ijerph17051729_

Round 1

Reviewer 1 Report

The authors address the limitations of their study, primarily the snowball sampling strategy which biases the sample. The explanation of limited resources and time sensitivity is a reasonable one, but it does limit their conclusions to a sub-population (53% of the sample are students, 88% of the sample are University educated) and makes the conclusions less generalizable to the entire population (particularly less educated people). 

This paper captures a snapshot of the psychological reaction very early in the course of a novel viral outbreak when information remains limited. It would be interesting to re-sample the same population after a few weeks before drawing any conclusions about the need for a focused public health initiative.

As 84.7% of the respondents had spent 20-24 hours per day at home in the initial phases of this epidemic, we can suppose that this isolation had a negative psychological impact on the respondents.

Perhaps the authors could try to explain the apparent disconnect between the IES-R (a PTSD scale) (N=651 (53.8%) rated moderate or severe psychological impact), and the DASS-21, (69.7% normal for depression, 63.6% normal for anxiety, and 67.9% normal for the stress sub scale).

Author Response

Replies to Reviewer 1 (Please note that all amendments were indicated by track changes)

Reviewer 1: The authors address the limitations of their study, primarily the snowball sampling strategy which biases the sample. The explanation of limited resources and time sensitivity is a reasonable one, but it does limit their conclusions to a sub-population (53% of the sample are students, 88% of the sample are University educated) and makes the conclusions less generalizable to the entire population (particularly less educated people). This paper captures a snapshot of the psychological reaction very early in the course of a novel viral outbreak when information remains limited. It would be interesting to re-sample the same population after a few weeks before drawing any conclusions about the need for a focused public health initiative.

Authors: Thank you for your comments and suggestion, Regarding your comment. “and makes the conclusions less generalizable to the entire population (particularly less educated people).” We have stated this as a limitation. While re-sampling is ideal, it is limited by the ethics requirement for collecting anonymous surveys without any contact information or personal identifiers to provide confidentiality of participants. As a result, we are not able to re-sample the same population in the future. We have added this as one of the limitations.

Furthermore, it would be ideal to conduct a prospective study on the same group of participants after a period. Due to ethical requirements on anonymity and confidentiality, we were not allowed to collect contact details and personal information from the respondents. As a result, we could not conduct a prospective study that would provide a more concrete finding to support the need for a focused public health initiative.

Reviewer 1: As 84.7% of the respondents had spent 20-24 hours per day at home in the initial phases of this epidemic, we can suppose that this isolation had a negative psychological impact on the respondents.

Authors: Based on results in Table 5, the number of hours spent at home per day did not have a statistically significant association with negative psychological impact, depression, anxiety and stress (p>0.05).

Reviewer 1: Perhaps the authors could try to explain the apparent disconnect between the IES-R (a PTSD scale) (N=651 (53.8%) rated moderate or severe psychological impact), and the DASS-21, (69.7% normal for depression, 63.6% normal for anxiety, and 67.9% normal for the stress subscale).

Authors: Thanks for highlighting this important point. The difference between IES-R and DASS-21 is due to the fact that IES-R assesses psychological impact after an event. In this study, respondents might refer to the COVID-19 outbreak as the event while DASS-21 did not specify an event. We have included this point under Discussion as follows:

The prevalence of moderate or severe psychological impact measured by IES-R was higher than prevalence of depression, anxiety and stress measured by DASS-21.  The difference between IES-R and DASS-21 is due to the fact that IES-R assesses psychological impact after an event. In this study, respondents might refer to the COVID-19 outbreak as the event while DASS-21 did not specify any event.

Reviewer 2 Report

The present manuscript addresses an important issue - the psychological impact of disease outbreak on impacted populations. As there is little literature on the COVID-19 virus at this time, the only improvement in background literature would be to discuss similar studies that may have been conducted for other disease outbreaks. 

For the procedure, the method of disseminating the survey and survey instrument are well explained. Was the survey conducted completely in English? If so, what are the associated limitations with an English language survey for the targeted population? What was the initial recruitment procedure before snowball sampling was instituted? Who was the survey first disseminated to and encouraged to pass on to others? Do you think the three day survey window was sufficient? How was this dissemination period selected?

There are minor English language related grammatical errors. These do not impact the overall flow of the paper. 

Author Response

Replies to Reviewer 2 (Please note that all amendments were indicated by track changes)

Reviewer 2: The present manuscript addresses an important issue - the psychological impact of disease outbreak on impacted populations. As there is little literature on the COVID-19 virus at this time, the only improvement in background literature would be to discuss similar studies that may have been conducted for other disease outbreaks. 

Authors: The fourth paragraph of the introduction is devoted to background literature on the psychological impact of an outbreak of infectious diseases. We cited 5 references (17 – 21) on the psychological impact of SARS and influenza outbreak on the mental health of the general public. Previously, most psychological research focused on health workers but there was less psychological research on the general public. 

Reviewer 2: For the procedure, the method of disseminating the survey and survey instrument are well explained. Was the survey conducted completely in English? If so, what are the associated limitations with an English language survey for the targeted population? What was the initial recruitment procedure before snowball sampling was instituted? Who was the survey first disseminated to and encouraged to pass on to others? Do you think the three-day survey window was sufficient? How was this dissemination period selected?

Authors: The survey was conducted in Chinese. We stated the language in the method. The online survey was first disseminated to university students and encouraged to pass onto others. We stated this under methodology. We have provided the number participants recruited each day.  The first day: 1120 participants; The second day: 86 participants; The third day: 4 participants. As the recruitment network was exhausted on the third day, prolonging the recruitment period would not cause a significant increase in the number of respondents. Furthermore, there is an urgency to publish the result as it may provide important information for healthcare workers and health authorities as the epidemic of COVID-19 is affecting more countries at this moment.

Reviewer 2: There are minor English language-related grammatical errors. These do not impact the overall flow of the paper.

Authors: We have conducted an additional grammar check. Two of the co-authors come from Singapore where English is their first language.